# Chronic Exposure to Lead and Cadmium in Residents Living near a Zinc Smelter

**DOI:** 10.3390/ijerph18041731

**Published:** 2021-02-10

**Authors:** HyeJeong Jo, GeunBae Kim, JunYoung Chang, Kwan Lee, ChulWoo Lee, BoEun Lee

**Affiliations:** 1Environmental Health Research Department, Environmental Health Research Division, National Institute of Environmental Research, Incheon 22689, Korea; hjjo0070@gmail.com (H.J.); mykgb@korea.kr (G.K.); acmecjy@korea.kr (J.C.); leecwoo@korea.kr (C.L.); 2Department of Preventive Medicine, Dongguk University, Kyungju 38066, Korea; kwaniya@dongguk.ac.kr

**Keywords:** zinc smelter, polluted area, environmental exposure, lead, cadmium

## Abstract

This study aimed to measure lead (Pb) and cadmium (Cd) exposure levels in residents living near a zinc (Zn) smelter in Seokpo-myeon, Bonghwa-gun, South Korea, and identify factors affecting exposure. Residents aged ≥20 years living within 3 km and ≥30 km away from the smelter were classified as the exposure group (n = 549), and the control group (n = 265), respectively. Data were obtained through a questionnaire survey. Blood Pb levels in the exposure group (4.19 µg/dL) were higher than in the control group (2.70 µg/dL). The exposure group (1.32 µg/L) also had higher urinary Cd concentrations than the control group (0.80 µg/L). Male sex, older age, previous work at the smelter, smoking, and proximity to the smelter were associated with higher blood Pb levels on multivariate analysis; urinary Cd concentration was significantly higher in women, those who were older, those with experience of working in a Zn smelter or mine, those with proximity to the Zn smelter, and those who consumed locally grown vegetables. In conclusion, Zn smelters are major source of Pb and Cd pollution and require ongoing public health management to prevent potential adverse health effects.

## 1. Introduction

Nonferrous metals exhibit remarkably diverse characteristics between the state of pure metals and near-infinite combinations of alloys, and they continue to show major developments. Smelting is a process whereby pure target metal is made from ore, during which various refining techniques are used to remove unwanted impurities in multiple stages. This process results in the release of harmful substances, such as lead (Pb), cadmium (Cd), nickel (Ni), and sulfuric acid, due to fugitive emissions from the target metal when melted at higher temperatures and from impurities or other ingredients.

Smelters are a well-known source of heavy metal emissions [1,2] and can affect the regional environment. One study from southern Mexico reported that heavy metal concentration in over 90% of roadside dust samples from regions near active smelters exceeded the Superfund cleanup goals, which are 5–65, 3–20 and 200–500 µg/g, for soil arsenic, cadmium and lead, respectively [3]. The composite pollution indices of heavy metals in surface soil within 1.5 km of a closed zinc (Zn) smelter in Magu Town, Guizhou Province, China, were 3–13 times higher than those of soils located more than 10 km away [4].

The fourth-largest zinc smelter in the world is the subject of this study and has been active since 1970 [5]. This smelter emits a large amount of heavy metal pollutants during the isolation of Zn from ore by electrolysis, the process of melting it in a furnace, and during the processing of by-products. In eight air samples from four locations in the smelter, the range of Pb and Cd concentrations were N.D–0.623 µg/m^3^ and N.D–0.059 µg/m^3^, respectively [6], showing that the Pb concentration from two samples was higher than the Korean ambient air Pb standard of 0.5 µg/m^3^. In a 2014 survey [7], the ambient air Cd concentration near the smelter was 0.0326 µg/m^3^, which exceeded the World Health Organization (WHO) standard (0.005 µg/m^3^) [8]. In soil samples from 448 locations within a 4-km radius around the smelter, the Korean soil pollution criteria for concern were exceeded for Pb, Cd, and Zn in 9, 59 locations, and 129 locations, respectively [6].

Environmental heavy metal pollution in the area around a smelter could have negative effects on the health of nearby residents. Heavy metals are particularly important from a public health perspective, as they can cause long-term toxic effects when they accumulate in organisms or the environment. Cd has a biological half-life of over 10 years [9] and is known to accumulate in bones and cause adverse health effects, including kidney dysfunction [10]. In one study of a population exposed to environmental pollution near a smelter in the United Kingdom (UK), urinary Cd concentration was significantly associated with indicators of renal damage [11]; similar results were reported in a recent study on residents near a closed copper (Cu) smelter in South Korea [12]. In a similar manner, chronic Pb exposure can negatively affect the blood, cardiovascular system, nervous system, and kidneys [13]. In a 25-year cohort study that included Pb smelter workers, standardized mortality rates due to cardiovascular disease, cerebrovascular disease, and chronic kidney disease were significantly increased [14]. Another study of residents near a Pb smelter in Dallas, USA, showed that chronic Pb exposure was associated with a deterioration in kidney function [15].

In the study area of the present study, the issue of environmental pollution due to the local smelter has been a persistent concern, and heavy metal pollution in the air, soil, and crops near the smelter has been reported in environmental surveys. As concerns have grown, it has become apparent that the health effects on residents of the surrounding area need to be investigated. In this study, we examined the heavy metal exposure levels and the contributing factors in residents living near the smelter, with the aim of providing scientific evidence to inform and support local environmental and public health management strategies.

## 2. Materials and Methods

### 2.1. Overview of the Study Area and Participants

The study area was a mountain village located in Seokpo-myeon, Bonghwa-gun, Gyeongsangbuk-do, South Korea. This region is especially vulnerable to environmental pollution due to its topography as it is a valley adjacent to a river that is surrounded by mountains (Figure 1) [6]. The Zn smelter near the residential area was established in 1970. It mostly produces Zn ingots, which are an important material in the steel manufacturing, automobiles, home electronics, and construction industries. The smelter produced a total of 350,000 tons of Zn in 2015. It also produces sulfuric acid (H_2_SO_4_), copper sulfate, silver granules, and indium. The smelter consists of three factories including electrolytic smelters and melting furnaces (three), storage tanks (two), H_2_SO_4_ tanks (four), and Zn concentrate storage buildings (two). In Korea, pollutant emission facilities such as smelters are managed under various environmental laws including air, water and soil. According to the Clean Air Conservation Act, smelters are monitored using telemonitoring systems inside the workplace and monitoring stations in surrounding areas of smelters.

A recent survey that investigated the area in which the soil environment was affected by particulate pollutants emitted from the smelter suggested that the directly-affected area was in a 1.5-km radius of the smelter; the indirectly affected area, however, which experienced an annual dust deposition rate of at least 30 kg/km^2^/year was within a 3-km radius of the smelter [6]. Based on this survey, we selected an area within 3 km of the smelter to represent the exposed region, which included parts of Seokpo-ri and Seungbu-ri in Seokpo-myeon. The exposure group consisted of 549 adults aged at least 20 years who lived within 3 km of the smelter. For the control group, we recruited 265 adults aged at least 20 years who lived in Murya-myeon, Bonghwa-gun, which was unaffected by the smelter due to it being located over 30 km to the west; this area has a similar population structure and the inhabitants have a similar socioeconomic status and lifestyle pattern to those living in the region of the exposure group. 

A total of 814 people were administered a questionnaire and also provided blood and urine samples to assess heavy metal exposure between 9 July 2016 and 17 July 2016, a time period that included two weekends.

The study was approved by the Dongguk University Gyeongju Hospital Institutional Review Board; the aims of the survey were explained to the participants and consent was received before proceeding with the survey.

### 2.2. Questionnaire

To develop the questionnaire, we reviewed environmental exposure assessment results from the study area [6] and previous studies on regions exposed to heavy metals, which were integrated following expert opinion. There were 35 main questions concerning the general characteristics of the participants (including sex, age, and duration of residence), environmental exposure (including occupational history and living environment), lifestyle habits (including smoking and alcohol consumption), disease history, and dietary habits (consumption of locally grown food).

The questionnaire part of the study took the form of one-to-one interviews conducted by a trained surveyor during health examinations. To account for participant accessibility, the survey was conducted at a community service center in Seokpo-ri for the exposed region and at the Murya-myeon town hall for the control region.

### 2.3. Sampling and Analytical Methods

Blood samples were collected using the vacuum tube method, and the urine samples collected were spot urine samples. The collected samples were frozen at −20 °C prior to analysis. The measured variables were blood Pb level and urinary Cd concentration. For blood Pb level measurement, 0.1 mL of blood was mixed with 0.3 mL of diluting solution (0.2% Triton X-100, 0.2% ammonium dihydrogen phosphate solution) and 0.1 mL of distilled water, and graphite furnace atomic absorption spectrometer (GF-AAS, PinAAcle 900, PerkinElmer, Waltham, MA, USA) equipped with the Zeeman background correction system was used to analyze the absorption at 283.3 nm.

For the urinary Cd concentration measurement, 0.2 mL of urine was mixed with 1.5 mL of diluting solution (0.2% Triton X-100, 0.2% ammonium dihydrogen phosphate) and 0.2 mL of nitric acid, and GF-AAS was used to analyze the absorption at 228.8 nm.

External quality assurance and control was accomplished with German External Quality Assessment Scheme (G-EQUAS). To ensure analytical quality, reference materials were obtained from the intercomparison program 56(G-EQUAS, 2015). Our values were 13.08, 40.19, 232.9, and 871.6 µg/L for four blood lead samples (reference values: 12.49, 37.64, 223.8, and 851.1 µg/L) and 0.20, 0.75, 7.8, and 13.5 µg/L for four urine cadmium samples (reference values: 0.23, 0.80, 8.4, and 13.1 µg/L), which were within the tolerance range appropriate for the confirmation of analytical methods reliability.

For the limit of detection, we used the method detection limit corresponding to 3.14 × the standard difference when seven samples of reference solution were analyzed. The values below the detection limit of all samples were 0.12% of blood Pb and 0.74% of urinary Cd. These values were replaced by 1/√2 of the limit of detection.

### 2.4. Statistical Analysis

The analysis of urine samples needs to deliberately consider the change in both urine concentrations and volume, which is difficult to accurately measure. Therefore, urinary biomarkers are corrected by creatinine excretion related to the urine excretion rate. However, with very dilute (urinary creatinine <0.3 g/L) or very concentrated (urinary creatinine >3.0 g/L) urine, exact results cannot be obtained [16]. There were 76 participants with urinary creatinine concentration outside the WHO-recommended range (0.3–3.0 g/L); they were excluded from the statistical analysis of urinary Cd concentration [17]. As the heavy metal concentrations showed skewed distributions, they were converted to the natural logarithm scale to normalize the distribution prior to statistical analysis. Participant ages were categorized as 20–49 years, 50–59 years, 60–69 years, or ≥70 years; the distance between the smelter and residence was categorized as ≤0.5 km, 0.6–1 km, 1.1–1.5 km, 1.6–3 km, and ≥30 km, and the duration of residence was categorized as either <10 years or ≥10 years. Chi-square tests were used to compare the sociodemographic characteristics, lifestyle habits, and dietary habits between the exposure and control groups. Student’s *t*-tests and analysis of variance were used to investigate differences in heavy metal concentrations in the body within each group depending on sociodemographic factors, health behaviors, and consumption of locally grown crops. The covariates in the multiple regression models were chosen as follows. First, the risk factors associated with exposure to blood Pb and urinary Cd were identified from the literature review. Second, the variable was selected as a candidate for multivariate analysis if related to heavy metal levels at *p* < 0.20 in univariate analyses. Based on these two perspectives, we determined the variables, which needed to be included in multivariate analysis as potential confounders.

The final model was selected using the forward stepwise method. Statistical significance was set at *p* < 0.05. SAS 9.4 (SAS Institute Inc., Cary, NC, USA) was used for all statistical analyses.

## 3. Results

The participants’ general characteristics are shown in Table 1. The sex distribution in the exposure group (men, 43.0%; women, 57.0%) was similar to that in the control group (men, 37.4%; women, 62.6%), but the mean age in the exposure group (60.3 years) was lower than that in the control group (67.7 years). The duration of residence in the current area differed significantly between the exposure (24.0 years) and control groups (33.7 years). The groups showed no differences in smoking status and alcohol consumption, but the proportion of participants eating locally grown beans, vegetables, and potatoes was higher in the control than in the exposure group. Meanwhile, because the reason for constructing the smelter in this area was due to the presence of the largest Pb and Zn mine in South Korea (Yeonhwa Mine, which closed in 1998), the percentage of participants with experience working in a smelter or mine was higher in the exposure (50.5%) than in the control group (4.2%).

Table 2 shows the heavy metal concentrations in the biological samples collected from the exposed and control groups. The mean blood Pb levels were significantly higher in the exposure group (4.19 µg/dL) than in the control group (2.70 µg/dL). The mean urinary Cd concentrations, both before and after adjusting for creatinine, were also significantly higher in the exposure group (1.32 µg/L, 1.47 µg/g-cr) than in the control group (0.84 µg/L, 1.14 µg/g-cr). These trends were maintained in a sensitivity analysis excluding participants with experience of working in a smelter or mine, revealing significantly higher blood Pb levels and urinary Cd concentrations in the exposure group. When the percentile distribution for heavy metal concentrations was examined, the 50th percentile was similar to the geometric mean. The 90th percentile blood Pb level in the exposure group exceeded the international recommended levels (5 µg/dL [US Centers for Disease Control and Prevention (CDC), 2012]), while the 90th percentile urinary Cd concentration in the exposure group was similar to the international recommended levels (4 µg/L [German, HBM II, 2010]).

When sex differences in heavy metal concentrations were analyzed in the exposure group, blood Pb levels were significantly higher in men, but both creatinine-adjusted and unadjusted urinary Cd concentrations were significantly higher in women (Table 3). Urinary Cd concentration showed a trend to increase with older age. There were no significant differences in blood Pb levels according to duration of residence, but urinary Cd concentrations were higher in participants who had lived in the area for ≥10 years compared to those who had lived for <10 years. In the exposure group, the participants with a history of working in a smelter or mine had significantly higher blood Pb levels, but not urinary Cd concentrations compared to those without such a work history. In terms of lifestyle habits, blood Pb levels were higher in the participants with a history of smoking and in non-drinkers, whereas both creatinine-adjusted and unadjusted urinary Cd concentrations were significantly higher in drinkers. The exposed region showed a much higher rate of consumption of locally grown foods than urban regions; thus, given that the problem of soil contamination has been highlighted in this region, we analyzed whether dietary habits affected the concentrations of heavy metals in the body. Both creatinine-adjusted and unadjusted urinary Cd concentrations were significantly higher in the participants that consumed locally grown beans, vegetables, or potatoes than those who consumed the equivalent store-bought foods. However, there were no significant differences in blood Pb levels between those who consumed locally grown and store-bought produce.

When we examined the results for the control group (Table 4), we observed differences in blood Pb and urinary Cd levels depending on sex, as well as differences in urinary Cd concentrations depending on age, similar to the results seen in the exposure group. Conversely, the control group participants showed a higher rate of consumption of locally grown produce compared to the exposure group, but they showed no significant differences in the heavy metal concentrations according to the source of their product.

When we analyzed the distribution of heavy metal concentrations in the body based on the distance between the smelter and a participant’s residence, blood Pb and urinary Cd levels were both significantly higher in the exposed region (within 3 km of the smelter) than in the control region (over 30 km from the smelter; Figure 2). Blood Pb concentration showed a trend to decrease with increasing distance from the smelter, but urinary Cd did not show a clear trend.

Table 5 shows the results of the multiple regression analysis used to identify factors affecting the participants’ blood and urinary heavy metal concentrations. The factors with significant effects on blood Pb levels were male sex, older age, experience of work in a smelter or mine, smoking, and living within 3 km of the smelter.

The factors with significant effects on urinary Cd concentration were sex, age, distance between the smelter and residence, and experience of work in a smelter or mine. Moreover, the consumption of locally grown vegetables also influenced urinary Cd concentration.

## 4. Discussion

Smelters are known to be a major source of heavy metal contamination [1]; the Zn smelter in the survey area of this study has been active for over 50 years, and heavy metal contamination due to smelter activity has been reported on several occasions. However, there are only a few surveys of heavy metal exposure to residents living near the smelter. Hence, in this study, we measured heavy metal concentrations in residents living near the smelter and investigated the factors affecting the exposure. 

In this study, the geometric mean of the blood Pb level in the residents of the exposed area was 4.19 µg/dL, with a range of 0.05–37.05 µg/dL; this was significantly higher than the level found in the residents of the control region (mean, 2.70; range, 1.07–6.77), and was also over that of Koreans of a similar age (≥60 years; 1.89 µg/dL) [18,19]. In the US National Health and Nutrition Examination Survey (NHANES) and the Canadian Health Measures Survey, the mean blood Pb levels were reported to be 0.82 µg/dL [20] and 0.93 µg/dL [21] in the general population, respectively. By comparison, the levels of those in the exposure group of our study were approximately 4 times higher. Even those with blood Pb levels below the previous US CDC reference value of 10 µg/dL have been shown to develop neurodevelopmental, cardiovascular, and immunological health effects. Accordingly, the recommendations were adjusted down to <5 µg/dL [22,23,24], while Germany rescinded the recommended value altogether [25]. In one Korean study of residents near a closed Cu smelter, the mean blood Pb level was 4.27 µg/dL, which was similar to that reported in the present study [26]. In contrast, a US study of residents near a Pb smelter reported a mean blood Pb level of 2.2 µg/dL, which was lower than our results [15]. Meanwhile, laborers occupational exposure to heavy metals showed higher levels than the residents in our study area; in the aforementioned US study, male workers at the Pb smelter showed blood Pb levels of 4.5 µg/dL [15], clerical workers in a Cu smelter in Japan showed levels of 5.5 µg/dL while smelting workers showed levels of 13.5 µg/dL [27], and workers at a nonferrous metal smelting plant in France showed levels of 38.7 µg/dL [28].

The geometric mean of the urinary Cd concentrations among residents in the exposed area in this study was 1.32 (range: 0.03–32.26) µg/L or 1.47 (range: 0.02–12.78) µg/g-cr; among the residents with no experience of working in a smelter or mine, the mean urinary Cd concentration was 1.36 µg//L (1.63 µg/g-cr). In terms of the international reference values for urinary Cd concentration, Germany’s Human Biomonitoring Commission II suggests a reference value of 4 µg/L [25]; 11% of the exposure group and 4% of the control group in our study showed urinary Cd levels that exceeded this international reference value. In biomonitoring studies on adults sampled from the general population, the mean urinary Cd concentration has been reported to be 0.34 µg/L among Koreans [18], 0.12 µg/L among Americans [20], 0.22 µg/L among Germans (50% tile) [29], and 0.16 µg/L among Canadians (50% tile) [21]. Cd concentration in the body is known to be higher in Asians, who, including Koreans, consume large amounts of grains compared to Westerners [30,31]. Thus, the urinary Cd concentration in the residents of the study area was higher than that of the general population, both domestically and internationally, and was also significantly higher than that of the control group (0.80 µg/L, 1.17 µg/g-cr). The geometric mean of urinary Cd concentrations in residents near a Zn smelter in Norway was 0.35 µg/g-cr [32], and the urinary Cd concentration in residents near a nickel-Cd battery factory in Sweden was 0.82 µg/g-cr for men and 0.66 µg/g-cr for women, which was lower than the results of our study [33]. However, in a study analyzing urinary Cd levels by the distance between a closed Cu smelter and a participant’s residence in Korea, the range of mean Cd concentrations was 1.88–3.62 µg/g-cr [12]; in China, residents near a Pb/Zn mine and a Cu smelter showed urinary Cd levels of 5.7 ± 3.1 µg/L and 5.5 ± 3.5 µg/L, respectively [34]. These concentrations were higher than those measured in our study.

Cd is known to promote nephrotoxicity. Long-term exposure to Cd can lead to its accumulation in the kidney and can cause neotubular dysfunction [35,36]. Additionally, Cd exposure increases the excretion of low molecular weight proteins in the urine, and when it increases by 10 times or more, it may cause nephropathy [37]; thus, we analyzed serum creatinine, glomerular filtration rate, and β_2_-microglobulin as indicators of renal function during the clinical examination. However, the proportion of participants showing a value outside the reference range for at least one of these variables was 18.4% (101 persons) in the exposure group and 15.8% (42 persons) in the control group, meaning that there was no statistically significant between-group difference (data not shown).

Sex, age, smoking status, and working history are factors affecting the concentration of heavy metals in the body. Previous studies identified the sex-specific difference in heavy metal metabolism enzymes and age-specific change in heavy metal distribution in the body [38,39], suggesting the increase in heavy metal exposure with smoking and occupational environments including smelters and mines [40]. Similarly, this study also confirmed that previously known factors affect the heavy metal concentrations in residents living near the Zn smelter. Even after adjusting for these factors in a multiple regression analysis, the distance from the participant’s residence to the smelter and the consumption of locally grown vegetables were significantly associated with blood Pb levels and urinary Cd concentration in this study.

Blood Pb levels were found to increase with older age. In a previous study, the geometric means of blood Pb in Koreans individuals were 1.25, 1.46, 1.59, 1.89, and 1.74 µg/dL for the following age groups: 19–29, 30–39, 40–49, 50–69, and ≥70 years old, respectively, showing an increasing trend with older age [19]. Similarly, a study by Muntner et al. reported higher blood Pb levels in the elderly (60–74 years old, 2.32 µg/dL vs. 18–37 years old, 1.28 µg/dL) [41]. Once absorbed into the body, Pb is excreted via the digestive system, but its biological half-life is approximately 10 years [42] and the distribution in the body is generally known to increase with age, up to around 50–60 years, after which it decreases again. This is explained as a cohort effect due to age and the half-life of metal ions in the body [43,44].

With aging, that the urinary Cd concentration increases along with the concentration of Cd in the kidneys [45]. In the present study, we observed a trend for higher urinary Cd concentration with older age, which is consistent with previous studies. One study investigating the general Korean population found increasing urinary Cd levels with older age [46], and another study of residents in parts of Korea without any obvious source of contamination showed that older age was associated with significantly increased urinary Cd concentration [47].

Blood Pb levels are known to be higher in men than in women; this is thought to be because men have more opportunities for social exposure to Pb based on their occupations and differences in lifestyle behaviors [48] and also because men have greater red blood cell volume, and large amounts of Pb are distributed in the red blood cells [49,50]. In this study, we observed significantly higher blood Pb levels in men rather than women, and average levels for the Korean population also indicate higher levels in men (1.87 µg/dL) than in women (1.37 µg/dL) [19]. Likewise, a study of residents living in an area at risk of environmental contamination near an iron manufacturing and petrochemical complex has reported higher blood Pb levels in men (2.95 µg/dL) than in women (2.20 µg/dL) [51]. Finally, in the NHANES study conducted between 1999 and 2002, the percentage of men with blood Pb concentration >5 µg/dL was 8.1% compared to just 2.2% in women [52].

Regarding Cd levels, women have been reported to absorb more Cd than men because they store less iron [53]; moreover, physiological factors have also been suggested to affect chemical toxicity in women, including changes related to menstruation, pregnancy, breastfeeding, and menopause [48]. In the present study, creatinine-adjusted and unadjusted urinary Cd concentrations were both higher in women, which is similar to the results of research conducted in the general Korean population [46] and in individuals living near a Pb/Zn smelter in the UK [11].

There are approximately 4,000 toxic or harmful substances in tobacco smoke, including Pb and other heavy metals [54]; smoking and tobacco smoke have both been reported to increase blood Pb levels [55]. We also observed higher blood Pb levels in smokers than non-smokers. This is consistent with the US NHANES study, in which smokers and past smokers showed elevated blood Pb levels compared to non-smokers [56], as well as studies in the Korean general population by Son et al. [57] and Joo et al. [46]. According to Kim et al., compared to non-smokers, past and current smokers have a 1.66- and 3.73-times higher risk of elevated blood Pb levels, respectively [58]; conversely, among residents in the exposed area, urinary Cd concentration was higher in non-smokers than in smokers. When participants were stratified by sex, urinary Cd concentration was higher in the non-smoking men compared to the smoking men, and higher in the smoking women compared to the non-smoking women. However, the percentage of smokers among men and women was 68.5% and 5.7%, respectively.

Compared to the participants living in an area far from the smelter (the control area), participants living near the smelter showed higher concentrations of heavy metals; this difference was even greater the shorter the distance from the smelter, which is similar to the results reported by Kim et al. [12]. One cohort study on pregnant women in Poland reported a decrease in blood Pb levels with increasing distance from a smelter acting as a source of Pb exposure, and this relationship was maintained even after adjusting for sociodemographic factors and lifestyle habits [59]. Moreover, a Chinese study on a region close to a Pb smelter reported higher Pb and Cd concentrations in the hair of people living closest to the smelter [60].

In the exposed area, the residents who consumed locally grown products had significantly higher urinary Cd concentrations than those who consumed store-bought vegetables; the source of food products was also identified as a factor affecting Cd concentration in the multiple regression analysis. Cd travels long distances in the air after binding to air particles and is also known to be absorbed by plants, especially leafy vegetables, and contaminates the food chain because it dissolves in water and binds strongly to the soil [61]. Green onions cultivated in the exposed area in this study showed Cd concentrations of 0.176–0.177 ppm, which exceeds the standard of 0.005 ppm [62]; in a soil survey, out of 448 samples collected from within a 4-km radius around the smelter, approximately 30% exceeded the heavy metal standards [63]. In the non-occupational exposure group, food is an important route of exposure to heavy metals [64]. Vegetables grown in contaminated soil show high levels of heavy metals, which can lead to potential health risks in people who consume them [65,66]. In a study of residents living close to a Zn smelter in Norway, urinary Cd concentrations were non-significantly higher in those who consumed fruits and vegetables grown at home [32]. Similarly, a study on people living near a Zn smelter in the UK did not observe any relationship between consumption of homegrown vegetables and urinary Cd concentration [11].

Several limitations affect the interpretation of the findings of this study. First, the proportion of the population that participated in the present study was modest. As of December 2015, there were 1553 residents aged at least 20 years living within the influenced area, of whom 549 persons (35.4%) participated in this study. When we compared the age and sex distributions between participants and non-participants, both men and women showed low participation rates in the 20–49 year group. This remains a limitation of this study despite our efforts to improve the participation rate by publicization through local broadcasts and banners and by providing convenience through weekend examinations and operating a taxi service between the residential area and the survey center. Given that we observed a trend for increasing heavy metal concentrations with older age, the low participation rate among participants under 50 years old, who have lower heavy metal levels, could have resulted in an overestimation of the levels of heavy metal exposure in the local residents. Second, unlike objective tools such as medical examinations, questionnaire surveys are more likely to involve local residents who participate voluntarily or are interested in the issues. Therefore, our data may be biased because there may have been highly valued indicators related to environmental exposure and health. Finally, our data may be subject to a statistical bias because the pairwise deletion method was applied for the management of missing data. However, as the amount of missing data in the present study was small, this bias might be considered small as well [67].

## 5. Conclusions

In this study, compared to a control area and the general population, we observed higher levels of heavy metals in residents living near a Zn smelter that is known to produce environmental heavy metal contamination in the soil and ambient air. Even after adjusting for potential confounding factors such as sex, age, smoking status, and occupational history, living near the smelter was demonstrated to be a factor affecting blood Pb levels and urinary Cd concentrations. Based on the results of this study, we propose that the local government should provide health monitoring for residents exposed to high levels of heavy metals. A follow-up monitoring project is underway, and continual environmental and public health management will need to be implemented to prevent potential health problems in the local population.

## Figures and Tables

**Figure 1 ijerph-18-01731-f001:**
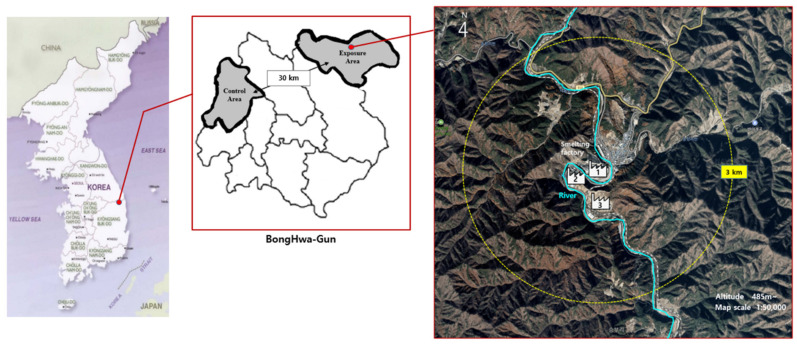
The survey area of this study.

**Figure 2 ijerph-18-01731-f002:**
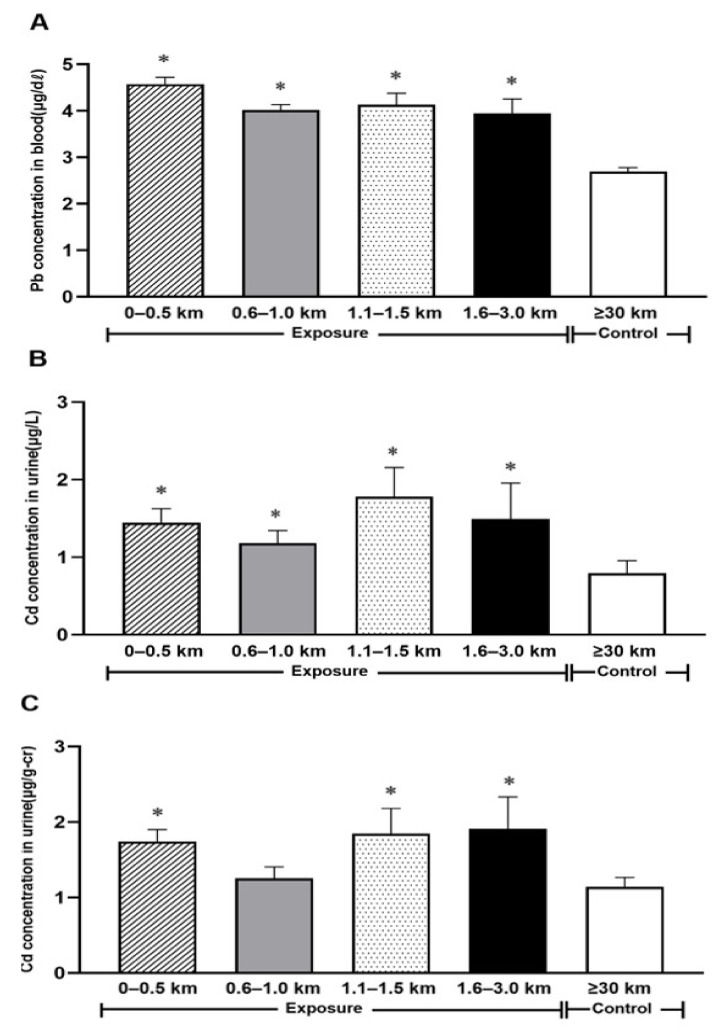
(**A**–**C**). Heavy metals concentrations according to distance between residence and zinc smelter. (**A**) Concentrations for Pb in blood. (**B**,**C**) Concentrations for Cd in urine. Each bar denote the geometric mean ± SE for heavy metals concentrations in biological samples (Exposure = Exposure group, Control = Control group). *: statistically significant difference from the control (*p* < 0.05).

**Table 1 ijerph-18-01731-t001:** General characteristics of the study participants.

Parameters	Exposure Group	Control Group	Total	*p*-Value
Total	549 (67.4%)	265 (32.6%)	814 (100%)	
Sex				
Men	236 (43.0%)	99 (37.4%)	335 (41.2%)	0.124
Women	313 (57.0%)	166 (62.6%)	479 (58.8%)	
Age (years)				
20 to 49	140 (25.5%)	22 (8.3%)	162 (19.9%)	<0.001
50 to 59	104 (18.9%)	37 (14.0%)	141 (17.3%)	
60 to 69	139 (25.3%)	75 (28.3%)	214 (26.3%)	
≥70	166 (30.2%)	131 (49.4%)	297 (36.5%)	
Mean ± SD	60.3 ± 15.0	67.7 ± 12.1	62.7 ± 14.5	<0.001
Duration of Residence				
< 10 years	167 (30.4%)	56 (21.1%)	223 (27.3%)	0.005
≥ 10 years	380 (69.2%)	209 (78.9%)	589 (72.3%)	
Missing data	3 (0.4%)	0 (0%)	3 (0.4%)	
Mean ± SD	24.0 ± 18.3	33.7 ± 22.1	27.1 ± 20.1	<0.001
Experience of Work in a Smelter or Mine				
No	270 (49.2%)	254 (95.8%)	524 (64.4%)	<0.001
Yes	277 (50.5%)	11 (4.2%)	288 (35.4%)	
Missing data	2 (0.3%)	0 (0%)	2 (0.2%)	
Smoking				
Non-smoker	372 (67.8%)	188 (70.9%)	560 (68.8%)	0.114
Smoker (~20 packs)	173 (31.5%)	77 (29.1%)	250 (30.7%)	
Missing data	4 (0.7%)	0 (0%)	4 (0.5%)	
Alcohol Drinking				
No	307 (55.9%)	132 (49.8%)	439 (53.9%)	0.414
Yes	239 (43.5%)	132 (49.8%)	371 (45.6%)	
Missing data	3 (0.6%)	1 (0.4%)	4 (0.5%)	
Beans				
Store-bought	389 (70.9%)	122 (46.0%)	511 (62.8%)	0.001
Locally grown	125 (22.8%)	139 (52.5%)	264 (32.4%)	
Not consumed	27 (4.9%)	3 (1.1%)	30 (3.7%)	
Missing data	8 (1.5%)	1 (0.4%)	9 (1.1%)	
Vegetables				
Store-bought	283 (51.5%)	79 (29.8%)	362 (44.5%)	<0.001
Locally grown	255 (46.4%)	183 (69.1%)	438 (53.8%)	
Not consumed	7 (1.3%)	2 (0.8%)	9 (1.1%)	
Missing data	4 (0.7%)	1 (0.4%)	5 (0.6%)	
Potatoes				
Store-bought	318 (57.9%)	90 (34.0%)	408 (50.1%)	<0.001
Locally grown	213 (38.8%)	174 (65.7%)	387 (47.5%)	
Not consumed	13 (2.4%)	1 (0.4%)	14 (1.7%)	
Missing data	5 (0.9%)	0 (0%)	5 (0.6%)	

SD: standard deviation.

**Table 2 ijerph-18-01731-t002:** Distribution of heavy metal concentrations in the exposure and control groups.

		GM ± GSD	Percentiles				
			10th	25th	50th	75th	90th
All participants	Exposure Group						
	B_Pb [μg/dL]	4.19 ± 1.87 *	1.96	2.88	4.20	6.22	8.68
	U_Cd [μg/L]	1.32 ± 2.63 *	0.39	0.73	1.43	2.47	4.12
	U_Cd [μg/g-cr]	1.47 ± 2.36 *	0.50	0.89	1.47	2.73	4.03
	Control Group						
	B_Pb [μg/dL]	2.70 ± 1.39	1.78	2.17	2.69	3.30	4.15
	U_Cd [μg/L]	0.80 ± 2.57	0.23	0.45	0.81	1.55	2.60
	U_Cd [μg/g-cr]	1.14 ± 1.86	0.52	0.75	1.17	1.73	2.44
Participants without occupational exposure (history)	Exposure Group						
	B_Pb [μg/dL]	3.47 ± 1.70 *	1.77	2.53	3.41	4.96	6.76
	U_Cd [μg/L]	1.36 ± 2.58 *	0.41	0.76	1.48	2.43	4.13
	U_Cd [μg/g-cr]	1.57 ± 2.33 *	0.55	0.94	1.69	2.85	4.06
	Control Group						
	B_Pb [μg/dL]	2.67 ± 1.39	1.78	2.14	2.64	3.27	4.09
	U_Cd [μg/L]	0.80 ± 2.60	0.22	0.45	0.82	1.61	2.61
	U_Cd [μg/g-cr]	1.16 ± 1.86	0.52	0.75	1.19	1.80	2.47

B: blood, U: urinary, Pb: lead, Cd: cadmium, GM: geometric mean, GSD: geometric standard deviation. * *p* < 0.01.

**Table 3 ijerph-18-01731-t003:** Concentration of heavy metals according to sociodemographic factors, health behavior, and consumption of locally grown crops in the exposure group.

	Number	B_Pb [μg/dL]	U_Cd [μg/L]	U_Cd [μg/g-cr]
Sex				
Men	236	5.31	1.08	1.05
Women	313	3.50	1.54	1.91
*p*-value		<0.001	<0.001	<0.001
Age (years)				
20 to 49	140	3.53	0.67	0.68
50 to 59	104	4.77	1.38	1.55
60 to 69	139	4.57	1.54	1.77
≥70	166	4.15	2.02	2.30
*p*-value		0.015	<0.001	<0.001
Duration of Residence				
<10 years	167	4.03	1.01	1.03
≥10 years	380	4.26	1.49	1.76
*p*-value		0.849	<0.001	<0.001
Experience of Work (in smelter or mine)				
No	270	3.47	1.36	1.03
Yes	277	5.03	1.29	1.72
*p*-value		<0.001	0.556	0.093
Smoking (~20 packs)				
Non-smoker	369	3.63	1.46	1.68
Smoker	177	5.66	1.07	1.13
*p*-value		<0.001	0.001	<0.001
Alcohol Drinking				
No	307	4.50	1.20	1.30
Yes	239	3.84	1.51	1.72
*p*-value		0.001	0.016	<0.001
Beans				
Store-bought	389	4.27	1.30	1.41
Locally grown	125	4.04	1.54	1.76
*p*-value		0.072	0.035	0.017
Vegetables				
Store-bought	283	4.04	1.18	1.32
Locally grown	255	4.38	1.54	1.70
*p*-value		0.420	0.001	0.001
Potatoes				
Store-bought	318	4.18	1.17	1.30
Locally grown	213	4.24	1.65	1.82
*p*-value		0.574	<0.001	<0.001

B: blood, U: urinary, Pb: lead, Cd: cadmium.

**Table 4 ijerph-18-01731-t004:** Concentration of heavy metals according to sociodemographic factors, health behavior, and consumption of locally grown produce in the control group.

	Number	B_Pb [μg/dL]	U_Cd [μg/L]	U_Cd [μg/g-cr]
Sex				
Male	99	3.06	0.73	0.82
Female	166	2.50	0.84	1.40
*p*-value		0.005	0.517	<0.001
Age (years)				
20 to 49	22	2.50	0.63	0.80
50 to 59	37	2.79	0.77	1.15
60 to 69	75	2.81	0.78	1.08
≥70	131	2.64	0.85	1.29
*p*-value		0.099	0.595	0.019
Duration of Residence				
<10 years	55	2.82	0.81	1.11
≥10 years	209	2.66	0.79	1.17
*p*-value		0.921	0.705	0.587
Experience of Work (in smelter or mine)				
No	254	2.67	0.80	1.15
Yes	11	3.35	0.80	0.88
*p*-value		0.162	0.806	0.145
Smoking (~20 packs)				
Non-smoker	188	2.58	0.75	1.21
Smoker	77	2.99	0.91	1.02
*p*-value		0.146	0.113	0.052
Alcohol Drinking				
No	132	2.94	0.81	1.08
Yes	132	2.48	0.79	1.23
*p*-value		0.012	0.545	0.063
Beans				
Store-bought	122	2.65	0.86	1.18
Locally grown	139	2.75	0.75	1.14
*p*-value		0.128	0.277	0.684
Vegetables				
Store-bought	79	2.67	0.84	1.15
Locally grown	183	2.71	0.79	1.17
*p*-value		0.572	0.745	0.838
Potatoes				
Store-bought	90	2.69	0.90	1.25
Locally grown	174	2.69	0.75	1.10
*p*-value		0.328	0.391	0.156

B: blood, U: urinary, Pb: lead, Cd: cadmium.

**Table 5 ijerph-18-01731-t005:** Forward stepwise multiple linear regression analysis for impact factors on level of heavy metals concentrations (log-transformed).

Heavy Metals	Exposure Factors	Unstandardized Beta	Standardized Beta	t	*p*-Value
B_Pb [μg/dL]	Sex (vs. male)	−0.163	−0.137	−3.204	0.001
	Age	0.005	0.121	3.611	<0.001
	Occupational history at smelter or mine (vs. no)	0.247	0.201	5.230	<0.001
	Smoking (vs. non-smoker)	0.173	0.136	3.361	0.001
	Alcohol drinking (vs. no)	−0.086	−0.072	−2.195	0.028
	Distance from smelter (vs. ≥30 km)				
	(vs. ≤0.5 km)	0.408	0.275	7.104	<0.001
	(vs. 0.6–1.0 km)	0.307	0.251	6.065	<0.001
	(vs. 1.1–1.5 km)	0.335	0.141	4.139	<0.001
	(vs. 1.6–3.0 km)	0.311	0.101	3.111	0.002
U_Cd [μg/g-cr]	Sex (vs. male)	0.571	0.359	10.712	<0.001
	Age	0.024	0.425	12.906	<0.001
	Occupational history at smelter or mine (vs. no)	0.159	0.098	2.601	0.009
	Distance from smelter (vs. ≥30 km)				
	(vs. ≤0.5 km)	0.515	0.263	6.714	<0.001
	(vs. 0.6–1.0 km)	0.345	0.212	5.047	<0.001
	(vs. 1.1–1.5 km)	0.500	0.159	4.729	<0.001
	(vs. 1.6–3.0 km)	0.317	0.074	2.279	0.023
	Locally grown vegetable (vs. buying)	0.103	0.065	2.057	0.040

B: blood, U: urinary, Pb: lead, Cd: cadmium.

## Data Availability

Data sharing is not applicable to this article.

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
