# Peer review of "Chronic Exposure to Lead and Cadmium in Residents Living near a Zinc Smelter"

_ijerph, 2021, doi:10.3390/ijerph18041731_

Round 1
Reviewer 1 Report
In the paper the authors analyze the influence of a zinc smelter on health exposure based on the detection of Pb and Cd levels in residents living within 3 km. The study is interesting, guiding and valuable. The paper could be accepted after minor revision. The specific comments are as followed,
(1) The authors should rewrite the abstract to avoid being lengthy and dilatory.
(2) The authors should mark clearly the sampling area including the exposure group and control group in the survey map.
(3) The authors should make up QA and QC in Sampling and Analytical Methods.
Reviewer 2 Report
This study measured lead and cadmium levels in residents living near a zinc smelter in Korea and identified possible influential factors. This paper is interesting; however, it needs major revision before being accepted. I have some major and minor comments.
Major points:
- Paragraphs 2-4 are too wordy. Authors need to trim those paragraphs to make them more concise.
- para 2: it is better to give examples in relevance to zinc smelters, rather than others (e.g., copper).
- section 2.1: Since the smelter may cause detrimental effects, it is better to add some background knowledge about the regulation from the government legislation aiming to achieve an effect control for the pollution.
- Section 2.1: How were the 814 people selected? Were the individual characteristics of the 814 people matched with the residents in the study areas? I’m worried about the generalizability of the selected sample.
- Section 2.3: please give the % for the measurements below the detection limits. Also, please specify how to deal with values below detection limits.
- Section 2.4: How to deal with the missing values in table 1? Multiple imputation is preferred before getting into the statistical analysis.
- Table 5: standardized beta should be given in order to detect which factor contributed most.
- limitation of the study should be discussed.
- para 4, p11 (Cd is known to promote…): the reason should be discussed.
- para 5, p11 (previously known factors…): Again, the reason should be discussed.
- the last paragraph in page 11 and the first paragraph in page12 primarily talk about the same thing. It is better to make them in one paragraph and refine it.
- para 4, p12: the last sentence (in the stepwise multiple regression analysis…) is irrelevant and should be removed.
Minor points:
- Authors need to add the line number.
- Language check is preferred.
- Table 2 should be adjusted, since it is not visually friendly.
Reviewer 3 Report
The manuscript is extremely interesting and the study was generally well-designed.
Minor comments:
Throughout the manuscript: The numbers in “m3” and “km2” should be superscript, and the numbers in chemical formulas such as H2SO4 should be subscript.
Line 44: Please specify what the “Superfund cleanup goals” are.
Lines 57-58: Actually, this is not true. Since the lower reported concentration was N.D., the Pb concentration was SOMETIMES higher than 0.5, but not always.
Lines 60-63: A decreasing trend is composed of at least three numbers, going from the highest to the lowest. In this case, the first reported value was the highest but the third value was not the lowest. Hence, saying that a decreasing trend was detected is incorrect.
Lines 65-69: A number of concentration exceedances only demonstrates that the soil was polluted. It does not say anything about the source of this pollution.
Line 138: Why the authors did not analyse the blood Cd level and the urinary Pb level? Are there any studies suggesting this methodology?
Lines 149-151: Please explain why the creatinine level is important and therefore the reason why those participants were excluded.
Lines 153-154: Why did the authors chose to put together such a wide range of ages (20-49), while the other ranges (50-59 and 60-69) were much smaller? Are there any previous studies suggesting this methodology?
Lines 155-156: Is this choice arbitrary or are there any previous studies suggesting this methodology?
Lines 160-165: This procedure is statistically wrong. A variable resulting univariately not significant might be significant if considered together with the other variables, due to the possibility of variable interactions.
Lines 184-188, 199-201, 203-207, 216-220 and 234-237: Please report the p-values each time a comparison between two groups of people is made. Alternatively, report in the methods section the significance level chosen for the whole manuscript.
Lines 286 and 292: Please remove one /.
Lines 306-308: This should also be reported in the Methods section.
Lines 376 and 378: “products” instead of “produces”
